# Analytical Models for Measuring the Mechanical Properties of Yeast

**DOI:** 10.3390/cells12151946

**Published:** 2023-07-27

**Authors:** Nikita Savin, Alexander Erofeev, Petr Gorelkin

**Affiliations:** Research Laboratory of Biophysics, National University of Science and Technology MISiS, Moscow 119049, Russia; peter.gorelkin@gmail.com

**Keywords:** SICM, AFM, micromanipulation, mechanical properties, cell stiffness, yeast, cell wall

## Abstract

The mechanical properties of yeast play an important role in many biological processes, such as cell division and growth, maintenance of internal pressure, and biofilm formation. In addition, the mechanical properties of cells can indicate the degree of damage caused by antifungal drugs, as the mechanical parameters of healthy and damaged cells are different. Over the past decades, atomic force microscopy (AFM) and micromanipulation have become the most widely used methods for evaluating the mechanical characteristics of microorganisms. In this case, the reliability of such an estimate depends on the choice of mathematical model. This review presents various analytical models developed in recent years for studying the mechanical properties of both cells and their individual structures. The main provisions of the applied approaches are described along with their limitations and advantages. Attention is paid to the innovative method of low-invasive nanomechanical mapping with scanning ion-conductance microscopy (SICM), which is currently starting to be successfully used in the discovery of novel drugs acting on the yeast cell wall and plasma membrane.

## 1. Introduction

A significant contribution to the determination of cell functions lies in the study of their structures, which in turn are characterized by mechanical properties. The elastic properties of cells change depending on their state. Therefore, the evaluation of these properties can be applied in cellular studies to obtain information about the state of health of cells or their level of damage [1]. There are two widely used approaches to the study of the mechanical properties of cells: (a) the study of the mechanical properties of the cell by an integral method, in which the cell is considered as a whole, and (b) the study of the mechanical properties of the structural components of the cell, in which lipid bilayers, biomembrane, and cytosolic proteins are studied in detail [2]. Probe microscopy methods are capable of visualizing and measuring the mechanical properties of soft or brittle biological objects both as a whole and in parts.

The classic method for studying the mechanical properties of yeast is cell rupture and the determination of the force required for this by micromanipulation [3]. At the moment, the most advanced method for studying the biomechanical properties of such cells is single probe microscopy (SPM), which has a high spatial resolution, allows research in biological media, and does not lead to cell death [4]. Significant progress has been made recently in the use of the atomic force microscopy (AFM) [5] and scanning ion-conductance microscopy (SICM) [6,7,8,9] methods to assess the physical properties of microorganisms. The main advantages of these SPM methods include the ability to conduct studies in physiological solutions, a small non-destructive mechanical effect on the biological object, and absence of the need for chemical fixation of the object, which makes it possible to study living cells. In addition, the high scanning speed (5 µm/s) and resolution (40 nm/pixel) make it possible to study the dynamics of nanoscale structures. Even though the resolution of AFM is higher than that of SICM, for living cells, the lateral resolution of SICM and AFM is comparable and is on the order of 10–20 nm. At the same time, the AFM method can mechanically affect the sample for local mechanical measurements, cell activation, or nanoscale “surgery”, while the SICM method can visualize fragile objects without contact [10]. As we have demonstrated earlier [6], it is impossible to visualize and measure the stiffness of drug-induced structural neoplasms on the surface of yeast using AFM.

In addition to visualization and obtaining mechanical properties, it is possible to record the physicochemical properties of samples via the AFM method when modifying the equipment. For example, using AFM-based force spectroscopy (AFM-FS), which makes it possible to study the forces of biomolecular adhesion at the level of individual molecules [11], one can obtain information about the intramolecular interactions of proteins and DNA/RNA chains [12] as well as analyze the molecular interactions of two single biomolecules attached to the substrate surface and AFM tip [13]. There are data on the implementation of the wavelet cross-correlation (XWT) technique in atomic force spectroscopy with simultaneous excitation of high cantilever modes to reproduce complex force dynamics with probe–sample contact [14]. XWT analysis provides parameters such as displacement, speed, and acceleration of the tip simultaneously for each contact; based on these data, it is possible to study the energy dissipation [15], which in turn provides information about the composition of the surface. Another application is nanoscale infrared AFM (AFM-nanoIR). It is based on the photothermal expansion of the absorbing regions of the sample by focusing the IR beam on the same region of the sample as the tip of the AFM, which causes the cantilever to oscillate in proportion to the aforementioned IR absorption [16]. AFM-nanoIR correlates topographic images from local areas of samples with chemical mapping. This can be used to determine the location of protein receptors on carcinogenic cells [17] or the interaction of amyloidogenic proteins with lipid layers [18]. Another combination that is gaining popularity is scanning electrochemical microscopy (SECM) with AFM or SICM. In these approaches, the probes are modified with a voltammetric ultramicroelectrode, making it possible to obtain the surface topography and electrochemical parameters. The method can be applied to various biological samples (living cells, yeast, bacteria, and DNA) as well as to record both extracellular and intracellular voltammograms [19,20].

It is worth noting alternative methods for extracting the mechanical properties of biological samples. For example, the recently presented parallel rheology model combines conventional viscoelastic elements with fractional calculus to record the macroscopic relaxation response of epithelial monolayers [21]. Because the model parameters are material properties, they are mechanical factors of the state of a biological object that can be used for diagnostics or as a target for regenerative medicine. Unfortunately, a significant limiting factor for the wide application of this model is the mathematical complexity of its fractional derivatives and the current lack of convenient numerical methods for analysis.

Another method for studying the elasticity of biological soft substances involves optical tweezers (OT). This method is based on a focused laser beam capable of influencing optical forces on micro- and nanoobjects due to momentum conservation during the interaction of light with matter [22]. Among its advantages are the non-contact method, the ability to study objects in the environment passing through the target, and a wide range of object sizes (from atomic [23] to micron size [24]) and cell types (living cells, bacteria, viruses, yeast [25,26,27,28]). Optical tweezers allow parameters such as the persistence length, Young’s modulus, and shear viscosity of liquid to be obtained; however, for visualization this method needs to be combined with other relevant methods, for example, multicolor epiluminescent fluorescence microscopy [29].

Unlike mammalian cells, microorganisms are endowed with a mechanically strong cell wall which is responsible for various cellular functions [30]. In this regard, the determination of the mechanical properties of the cell wall and the factors affecting them is important in the study of yeast or bacteria [31]. However, to extract useful information about the properties of the cell wall, the obtained experimental data must be mathematically modeled. The choice of a mathematical model is a decisive factor in the analysis of the obtained experimental data, as the plausibility of the determined physical quantity depends on it. At the moment, for the methods of probe microscopy and micromanipulation there are many different analytical models applicable to microorganisms. In this regard, the purpose of this review is to present the currently available mathematical models for various methods of studying the mechanical properties of yeast. We caution that the reader should refer to the original publication for the use of the presented equations. The equations presented in this paper are needed to verify whether the experiments conducted by the researchers contain the input parameters of the functions.

In the future, this review may contribute to the selection of optimally suitable functions of the energy of deformation, compression, or tension to create a new technique for obtaining the elastic modulus using the SICM method. At the same time, the combination of models of individual components of an animal cell with yeast models can expand the application of the SICM technique in study of the mechanical properties of both healthy yeast and yeast subjected to biological action, which can lead to the formation of soft structures on the cell surface with a modulus of elasticity of about kPa.

## 2. Mathematical Models for Measuring Mechanical Properties by Micromanipulation

One of the common methods for characterizing the mechanical properties of cells is micromanipulation compression [32,33,34]. In such experiments, force–deformation data are obtained by compressing the entire cage between flat parallel plates and measuring the force acting on it. As a result, the data obtained can be mathematically modeled to provide information about the mechanical properties of the cell wall. The mathematical models considered in this chapter are shown in Table 1, and their schemes are provided in Figure 1.

A simple model used in the micromanipulation method and suitable for the structure of yeast cells is the hollow sphere model. There are works that consider the modeling of compression of hollow spheres based on spherical shells filled with gas or spherical shells filled with an incompressible liquid [36,39]. They assume that the wall of the sphere is thin enough to be considered as a thin elastic shell in which stresses are expressed as wall tension and the wall cannot withstand out-of-plane shear stresses or bending moments. Because non-zero stresses are in the plane of the cell wall (axes OY and OX), this position is described as a plane stress (the principal stress along the OZ axis is assumed to be zero). In this case, the volume of the liquid is assumed to be constant, which indicates the impermeability of the sphere wall. An alternative finite element analysis approach based on the sea urchin model [40] was described and applied in [41], and the finite element method was applied with the assumption of cell wall permeability in [42], specifically, a variable volume of liquid inside the sphere.

The analysis of force–deformation data in compression can be carried out using the equation relating the stresses and deformations of the cell wall. Assuming that the cell wall is hyperplastic, the stress components can be derived from the strain energy function. There are different variants of the strain energy function applicable to cell walls. Examples of such functions are briefly provided in the work of J.D. Stanson et al., where they were used to interpret data on cell compression [35]. For example, certain models assume that the wall is incompressible [39,43], while others consider the strain energy function, which includes the Poisson ratio (for yeast, the Poisson’s ratio is assumed to be 0.5) [40,42]. When the constitutive equation or strain energy function is adopted, fitting the experimental data should allow the characterization of the properties of the cell wall material. However, to obtain an unambiguous value of the modulus of elasticity it is necessary to accurately measure such geometric parameters as the initial degree of stretching, the thickness of the cell wall, and the loss of fluid volume [42]. After this, the final equation must be evaluated using various types of mechanical tests.

For example, in the work of J.D. Stanson et al. an analytical model was developed and experimentally confirmed to describe the compression of a single yeast cell between parallel flat surfaces [35]. Based on the strain energy function of the sea urchin egg model (Figure 1A), the strain energy function of the cell wall is expressed as follows [40]:(1)$$W=\frac{Eh_{0}}{2\left(1-v^{2}\right)}(\varepsilon_{1}^{2}+\varepsilon_{2}^{2}+2v\varepsilon_{1}\varepsilon_{2})$$
where *W* is the strain energy per unit of initial volume, ε is the infinitesimal strain, (with ε_1_ being the index meridional direction and ε_2_ the longitudinal direction), *E* is the modulus of elasticity, *h*_0_ is the initial wall thickness, and *v* is Poisson’s ratio.

Based on the assumption that this is the infinitesimal strain limit of the general large strain equivalent, which is assumed to be applicable to all deformations, the authors of the above work extended Equation (1) to large deformations, replacing infinitesimal strain directly with the Green strain (*E_i_*)
(2)$$W=\frac{E}{2\left(1-v^{2}\right)}(E_{1}^{2}+E_{2}^{2}+2vE_{1}E_{2})$$
and Hencky strain (*H_i_*)
(3)$$W=\frac{E}{2\left(1-v^{2}\right)}(H_{1}^{2}+H_{2}^{2}+2vH_{1}H_{2})$$

In another publication, Feng and Yang described the compression of a spherical membrane filled with liquid (Equations (4)–(7), Figure 1B) [36]. The cell wall in this model is divided into areas in contact with and not in contact with the compressive regions. There are separate groups of defining equations for the contact and non-contact regions:(4)$$\frac{{d\lambda }_{1}}{d\psi }=-\frac{\lambda_{1}}{\lambda_{2}sin\psi }\left(\frac{f_{3}}{f_{1}}\right)-\left(\frac{\lambda_{1}-\lambda_{2}cos\psi }{sin\psi }\right)\left(\frac{f_{2}}{f_{1}}\right);$$
(5)$$\frac{{d\lambda }_{2}}{d\psi }=\frac{\lambda_{1}-\lambda_{2}cos\psi }{sin\psi };$$
(6)$$\frac{{d\lambda }_{1}}{d\psi }=\left(\frac{\delta cos\psi -\omega sin\psi }{{sin}^{2}\psi }\right)\left(\frac{f_{2}}{f_{1}}\right)-\left(\frac{\omega }{\delta }\right)(\frac{f_{3}}{f_{1}});$$
(7)$$\frac{{d\lambda }_{2}}{d\psi }=\left(\frac{\omega sin\psi -\delta cos\psi }{{sin}^{2}\psi }\right);$$
where ψ is the angular position of the point measured from the vertical axis of symmetry, *f_i_* represents the functions of the principal tensions, and *δ* = λ_2_sin*ψ*; and ω = *dδ*/*dψ*.

Cell compression consists of axisymmetric deformation; thus, λ_1_ and λ_2_ are principal stretches in the meridian and circumferential directions. The last modeling step is to use the strain energy functions from Equations (1)–(3) in Equations (4)–(7) to obtain the stresses.

Another approach was proposed in [44] as a simplification of an earlier model [34] which adopted the method of infinitesimal deformations. In this model, a cell is considered a fluid-filled sphere with thin, compressible, and linear elastic walls. Any change in the initial uninflated cell wall thickness *h*_0_ and the final cell wall thickness *h* is considered negligible. The principal Cauchy stresses in the wall are expressed as follows:(8)$$T_{1}=\frac{Eh_{0}}{1-v^{2}}\{\left(\lambda_{1}-1\right)+v(\lambda_{2}-1)\}$$
(9)$$T_{2}=\frac{Eh_{0}}{1-v^{2}}\{\left(\lambda_{2}-1\right)+v(\lambda_{1}-1)\}$$
where λ_1_ and λ_2_ are the principal stretch ratios, equal to the ratio of the length of the membrane section before and after deformation; indices 1 and 2 correspond to the meridional and circumferential directions, respectively; *T*_1_ and *T*_2_ are the tensions in the wall corresponding to λ_1_ and λ_2_, respectively; *E* is the Young’s modulus of the cell wall; *v* is Poisson’s ratio; and *h*_0_ is the initial thickness of the cell wall.

In their model [35], Stanson et al. introduced the functions of infinitesimal strain (Equations (10)–(12)) based on the method of infinitesimal deformations [44]. Through the generated strain energy functions in terms of the second Piola–Kirchhoff stress and using the definition of the Green strain, the authors determined the main stresses in the case of finite strain in Equations (13)–(15) [45]. Using the definition of Hencky strain, the main stresses are expressed in Equations (16)–(18).

Infinitesimal strain:(10)$$f_{1}=\frac{Eh_{0}}{1-v^{2}}$$
(11)$$f_{2}=\frac{vEh_{0}}{1-v^{2}}$$
(12)$$f_{3}=\frac{Eh_{0}(\lambda_{1}-\lambda_{2})}{1+v}$$

Finite (Green) strain:(13)$$f_{1}=\frac{Eh_{0}}{2\left(1-v^{2}\right)}\{3\lambda_{1}^{2}+v\lambda_{2}^{2}-(1+v)\}$$
(14)$$f_{2}=\frac{Eh_{0}}{2\left(1-v^{2}\right)}\frac{\lambda_{1}}{\lambda_{2}^{2}}\left\{v\lambda_{2}^{2}+\lambda_{1}^{2}+\left(1+v\right)\right\}$$
(15)$$f_{3}=\frac{Eh_{0}}{2\left(1-v^{2}\right)}\{\frac{\lambda_{1}}{\lambda_{2}}[\lambda_{1}^{2}-\left(1+v\right)-\frac{\lambda_{2}}{\lambda_{1}}[\lambda_{2}^{2}-(1+v)\}$$

Hencky strain:(16)$$f_{1}=\frac{2Eh_{0}}{{3\lambda_{1}^{2}\lambda }_{2}}\{2-\ln\left(\lambda_{1}^{2}\lambda_{2}\right)\}$$
(17)$$f_{2}=\frac{2Eh_{0}}{{3\lambda }_{1}\lambda_{2}^{2}}\{1-\ln\left(\lambda_{1}^{2}\lambda_{2}\right)\}$$
(18)$$f_{3}=\frac{2Eh_{0}}{{3\lambda }_{1}\lambda_{2}}ln\{\frac{\lambda_{1}}{\lambda_{2}}\}$$

By fitting the force–strain curves obtained through numerical simulation to experimental data, Stenson et al. determined the elastic modulus and the initial stretching coefficient [35]. The experimental data obtained by the authors are consistent with their mechanical model of the cell wall; however, the desired characteristics depend on the Poisson’s ratio and the thickness of the cell wall. In their subsequent work, Stenson et al. conducted an experiment on compression using micromanipulation at already high strain rates [31]. The data of the force–strain curves for yeast cells confirmed the operability of the previously used mechanical model of the cell and the possibility of neglecting the permeability of the cell wall at high strain rates, which made it possible to find the initial coefficient of cell expansion with the appropriate modulus of elasticity. In addition, the value of the elastic modulus agrees with the AFM data [46]. In this case, fixing the initial stretching coefficient leads to an inaccurate estimate of the elastic modulus.

Cellular expansive growth, typical of fungal cells, has mechanical aspects that are important in creating theoretical, mechanical, and biophysical models. Based on the work of A. Geitmann et al. [47] on the mechanics and modeling of cell wall growth and V. Fliert [48] on the expansion of thin viscous shells, Banavar et al. derived equations that determine the dynamics of the growing cell wall. The local normal cell wall force balance is [37]:(19)$$\sigma_{ss}k_{s}+\sigma_{\varphi \varphi }k_{\varphi }=P;\sigma_{ss}k_{s}=P/2,$$
where *s* is the parametrized arc length from the projection vertex, *φ* is the azimuth angle, *k*_s_ and *k*_φ_ are differential equations with respect to s and φ, *P* is the internal turgor pressure of the cell, and *σ*_ss_(s,t) and *σ*_φφ_(s,t) are stresses along s and φ in the cell wall.

The response of a cell wall to stresses applied to it depends on its mechanical properties. It should be considered that the yeast cell wall is elastic for short periods of time [49] but expands irreversibly during division [50], with a characteristic liquid-like behavior of the cell wall in the growth areas. Therefore, Banavar et al. [37] suggested that the growing cell wall behaves as an inhomogeneous viscous liquid with a spatially changing viscosity *µ*(*s*) that increases with distance from the growth apex (Figure 1C). The local tangential velocity *u*(*s*,*t*) of a cell wall with a constant density is equivalent to its deformation rate *ὲ_s_* = *δu/δs* as well as to *ὲ_φ_* = (1/*r*)(*dr*/*dt*), where *r* is the local radius of the cell wall, and is related to stresses in the cell wall by Equations (20) and (21) [48]:(20)$$\sigma_{ss}=4\mu h\left[{ὲ}_{s}+\frac{{ὲ}_{\varphi }}{2}\right];$$
(21)$$\sigma_{\varphi \varphi }=4\mu h\left[\frac{{ὲ}_{s}}{2}+{ὲ}_{\varphi }\right],$$
where h is the cell wall thickness.

As a result, we can summarize the main points that should be considered when developing an analytical model. First, the uniqueness of the obtained modulus of elasticity depends on the accuracy of measuring the parameters of the cell wall thickness, the initial degree of stretching, and the loss of fluid volume. Second, at high strain rates it is possible to neglect the permeability of the cell wall and use the impermeable sphere model. Third, not all parts of the cell are always elastic, meaning that during cell growth the behavior of the cell wall has a fluid-like character.

## 3. Mathematical Models for Measuring Mechanical Properties by the AFM Method

Atomic force microscopy is a well-established method that allows the study of various parameters of biological objects, including the mechanical properties of cells. The basic principle of AFM operation is based on recording the force interaction between a probe and the sample surface [51]. The probe is a tip mounted on a flexible cantilever that is illuminated by a laser beam. When the tip is mechanically deflected from the sample, the laser beam deflection is fixed by a photodiode, the magnitude of which depends on the forces of interaction between the tip and the sample (Figure 2). During the existence of this method many analytical models have been developed, the most relevant of which are presented in this section (Table 2). Model schemes are shown in Figure 3.

The Hertz model is used in most works devoted to evaluating the Young’s modulus of cells by AFM; this model describes a simple case of elastic deformation of two ideally homogeneous smooth bodies in contact under load (Figure 3A) [52]. The following assumptions are used in the model: (a) the shape of the indenter is parabolic, and (b) the sample thickness is much greater than the indentation depth.

The force exerted by a cantilever with a probe of radius *R* on the cell is determined by the equation
(22)$$F\left(h\right)=\frac{4\sqrt{R}}{3}Eh^{\frac{3}{2}}$$
where *h* is the indentation depth and *E** is the effective modulus of the probe–sample system, which is calculated by the equation:(23)$$\frac{1}{E^{*}}=\frac{1-v_{tip}^{2}}{E_{tip}}E^{*}h^{\frac{3}{2}}$$
where *E_tip_*, *v_tip_*, *E_sample_*, and *v_sample_* are the Young’s moduli and Poisson’s ratios for the tip and sample materials, respectively. In the case where the tip material is much harder than the sample material, the following equation is used [64]:(24)$$E^{*}\approx \;\frac{E_{sample}}{1-v_{sample}^{2}}$$

The Hertz theory does not allow the probe to stick to the sample; for this case, Kendall and Roberts modified the model [65]; to characterize the elastic properties of cells, Mahaffy et al. used the dynamic Young’s modulus [66]. However, the cell surface is heterogeneous, and there is an error in the evaluation of the cell’s Young’s modulus by the Hertz model. In 1975, Derjaguin et al. presented the Deryagin–Muller–Toporov (DMT) model, which includes cohesive forces outside the contact surface (Figure 3B) [53]. In their theory, the profile of the deformed surface corresponds to the Hertz model. The DMT model is applicable in the presence of long-range surface forces outside the region of contact between the probe and the sample, and is valid when there is weak adhesion between the nanoindenter and the outer surface of the sample. Ricardo et al. used the DMT model to determine the AFM Young’s modulus of yeast cells as well as the change in cell wall stiffness under chemical stress caused by acetic acid [67,68]. It should be noted that in the DMT model, due to geometric limitations the real contact area and the depth of the dent are considered negligible. In this connection, the use of the model is a priority for objects with low cohesion and a small radius of curvature.

Another model for studying the elastic properties of cells by AFM is a model based on the theory of elastic shells [55,56], in which cells are represented as shells filled with liquid. In this model, the effective Young’s modulus is estimated from the ratio between the effective Young’s modulus, the shell thickness, and the bending modulus. The main problem with this approach is the determination of the boundary conditions and the contact radius of the probe with the sample. In addition, when evaluating the mechanical characteristics by the AFM method, the finite element model can be used [69].

Another way to model the mechanical characteristics of yeast is to represent the cell as a cylindrical shell. To test the finite element model, Zhao et al. [46] compared simulation results with theoretical results in cases where the shell is inflated [56] and indented (Figure 3C) [70]. In the case when the edges of the shell are free from restrictions, the internal pressure *p* creates only a hoop stress, and the radius of the cylinder increases by [56]
(25)$$\delta =\frac{pR^{2}}{Eh},$$
where *δ* is the radial displacement, *R* is the cell radius, *h* is the wall thickness, and *E* is the modulus of elasticity.

In the case where the cylinder is subjected to equal and opposite radial loads, the equation for radial indentation is [70]
(26)$$\delta (x,ϴ)=\frac{F}{Eh}f(\frac{R}{h},x,\theta ),$$
where *f* is a complex function of *R/h* and the coordinate parameters *x* and *θ*.

Assuming that the cell wall is incompressible, L. Zhao et al. [46] modeled the mechanical model using the finite element model. At the end of each simulation, the maximum radial displacement is obtained with predetermined values of the elastic modulus *E*, the force *F* applied to the cage, the cage radius *R*, and the wall thickness *h*. The authors found that *F* and *δ* are linearly dependent on each other, while the cell wall elasticity constant *k_w_* depends on the mechanical properties and dimensions of the cell wall but does not depend on the internal pressure of the cell. By transforming the correlation, they calculated the elastic modulus of the cell wall:(27)$$E=0.8\frac{k_{w}}{h}{\left(\frac{R}{h}\right)}^{1.5}$$

The main problem in studying the mechanical properties of the yeast cell wall is the frequent discrepancy between the elastic modulus obtained by AFM and micromanipulation methods. Thus, according to the data obtained by the AFM method, the Young’s modulus is about 0.2–1.6 MPa [71,72,73], while in studies using the micromanipulation method the value is about 100–200 MPa [31,74]. The reason for this discrepancy is the use of the classical Hertz–Sneddon analysis in AFM measurements, where it is assumed that the entire cell is a single continuous material [75]. The cell membrane is considered unstressed, and the force resisting deformation is negligible compared to the force of reaction from the cytoskeleton [76]. Therefore, AFM experiments analyzed using the Hertz–Sneddon equations provide information about the mechanical properties of the cytoskeleton, not the cell membrane. Because the cell wall is much stiffer than the cytoskeleton, the Young’s modulus of the cell wall obtained using the Hertz–Sneddon analysis has no physical meaning.

Vella et al. [54] considered the internal pressure in yeast cells and, using the elastic shell model (Figure 1D), calculated the Young’s modulus from the published AFM data of yeast cells [77], obtaining corrected values of 12–46 MPa, which is an order of magnitude higher than the values obtained using the Hertz model. However, their work is based on the theory of shells, in which the thickness of the shell h is much smaller compared to the radius of the cell sphere. Mercade’-Prieto et al. considered the case where the thickness of the yeast cell wall is large [38] and its analysis using the shell theory is impossible. They considered the indentation of spheres using finite element method (FEM) modeling, which considers Green strains and Hencky strains.

Using FEM during compression of a spherical shell (*h/r* = 0.05) with a sharp indenter (*r_ind_/r* = 0.01) in the case of a single-layer cell wall, Mercade’-Prieto et al. confirmed [38] that the Reissner equation (Equation (28)) is applicable only to thin shells (*h/r* < 0.02); in the case of thicker shells, in which the probe is pressed into the shell, the Hertz–Sneddon equations are valid [78]. The values of the point loading *F* and wall stiffness *Eh* (where h is the wall thickness) in this case are calculated according to Equations (28) and (29):(28)$$F=\frac{4Eh^{2}}{\sqrt{3(1-v^{2})}}\frac{d}{r};$$
(29)$$Eh=\frac{2k_{wd}}{f(h/r)},$$
where *f*(*h/r*) is a function depending on the deformation model and *k_wd_* is the cell wall elasticity constant.

In studies by Mercade’-Prieto et al., the corrected value of the Young’s modulus is 10 MPa [38], which is higher than when using the Hertz–Sneddon analysis but lower than when using micromanipulation compression [31].

The authors then considered a double-layer model (Figure 1D) in which it is assumed that the outer layer (shell) of the cell wall has a thickness *h_out_* with a Young’s modulus *E_out_* and is attached to an inner layer (core) of the cell wall which has a thickness *h_in_* with a Young’s modulus *E_in_*. For the micromanipulation method, in the case of compression of a two-layer core-shell sphere to large deformations, the Young’s modulus is calculated by Equation (30). The results of the FEM are consistent with the micromanipulation data, and the Young’s modulus *E_in_* is about 0.4–0.8 GPa.
(30)$$E_{total}={\left(Eh\right)}_{in}+{\left(Eh\right)}_{out}$$

When using the AFM method and a sharp indenter, the behavior of a two-layer cell wall at small deformations depends on the thicknesses of the two layers and the relative values of their elastic moduli *E_in_*/*E_out_*. When *E_in_*/*E_out_* = 1, the system behaves as a single-layer wall; when *E_in_*/*E_out_* = ∞, however, the system is analogous to the cell wall being pressed against a very rigid substrate [79]. In this regard, it is incorrect to use the Hertz–Sneddon analysis when *E_in_*/*E_out_* is low, while its use is optimal when *E_in_*/*E_out_* is high. FEM results obtained by Mercade’-Prieto et al. using the double layer model were determined using the pseudo-Hertz Equation (31) with pseudo-Hertz Young’s modulus (*E_pH_*) and Hencky strain ε = 0.004, in which the vertex displacement (*d*) is equal to the indentation depth (*d_ind_*) [38].
(31)$$F_{paraboloid\;}=\frac{4r_{ind}^{0.5}(2r{)}^{1.5}}{3(1-v^{2})}E_{pH}\varepsilon^{1.5}$$

Equation (31) describes the force profile only for small deformations (*d* < 0.2*h_out_*). At a higher value of *E_in_*/*E_out_*, the inner layer behaves as a rigid substrate, in which case *d* ≈ *d_ind_* and *E_pH_*~*E_out_*. It is worth considering that when a rigid inner layer is present the outer layer is highly deformed even with low indentation. Thus, the model of a two-layer cell wall suggests the possibility of estimating the elasticity modulus by the AFM method (with a sharp probe) of only the outer layer and using the micromanipulation method to estimate the total stiffness of the wall. It should be noted that the determination of the Young’s modulus is affected by the contribution of the inner layer and by the rigidity of the substrate.

From models of contact mechanics based on the theories of Hertz, Boussinesq, and Sneddon [80,81,82], analytical expressions can be derived that relate the Young’s modulus, indentation, and force [81,82,83]. The main assumption in these models is the consideration of the sample as a layer of infinite thickness. In models of semi-infinite contact mechanics the rigidity of the substrate is not considered, which can lead to errors in determining the Young’s modulus of the cell. This is due to the reflection of the voltage applied by the probe back to the cell surface. This effect has been described as an artifact of the bottom effect [84].

P. Garcia and R. Garcia presented a method for determining the elasticity modulus of mammalian cells attached to a solid substrate (Figure 3E) [57]. The authors used a non-Hertz model to express the change in the area of contact with the imprint. In the presented theory, the force is defined as a function of the imprint and the contact radius as the sum of terms expressed in reciprocal powers of the sample thickness. In the case of a paraboloid probe with a shape calculated according to Equation (32), the force is expressed by Equation (33), while the expression is applicable only when the indentation is less than or equal to the radius of the tip:(32)$$f\left(r\right)=\frac{r^{2}}{2R}$$
(33)Fspere=F01h0+1.133δRh+1.497δRh2+1.469δRδRh3+0.755(δ2R2)h4,F0=169EcellRδ32
where radial coordinate *r* = ((x^2^) + (y^2^))^1/2^, *R* is the radius of the probe, h is the height of the sample, *δ* is the indentation, *F*_0_ is the applied force, and *E_cell_* is the Young’s modulus of the cell.

By performing finite element simulations, the authors confirmed that the Young’s modulus of cells measured by AFM depends on the solid substrate and that the artifact of the bottom effect is determined by the ratio between the contact radius and the thickness of the cell. The bottom effect theory of elasticity describes the above features, allowing the determination of the true elastic modulus of the cell without the influence of a solid substrate. However, there are cases in which the inaccuracy in determining cell rigidity is not caused by the bottom effect but by the presence of the cortical layer of the cell, which is much more rigid than the underlying internal cytoskeleton.

R. Vargas-Pinto et al. used the example of endothelial cells of the human umbilical vein to determine the reason for the discrepancy between the data on cell elasticity obtained for sharp and spherical probe tips (Figure 3F) [58]. Using FEM, the authors modeled the tip of the probe as a solid body [85], then the geometry of the sharp tip was simplified from a pyramid to a cone. The cell cytoskeleton was modeled as a cylindrical disk with a given radius and thickness and with a cortical layer surrounding the cell of a given thickness. The authors calculated the values for a spherical tip using Equation (22), while in the case of a sharp tip they used the previously described models of Rico et al. [59] and Briscoe et al. [60] for pyramidal and conical indenters, respectively, where it is taken into account that the sharp pointed tips are not an ideal cone but have a spherical cap at the top, meaning that the model behaves as a spherical tip at small indentations (*δ* < *b*^2^/*R*) and as a conical tip for large deformations:(34)$$F=\frac{2E}{1-v^{2}}\left(a\delta -m\frac{a^{2}}{\tan\theta }\left[\frac{\pi }{2}-{sin}^{-1}\left(\frac{b}{a}\right)\right]-\frac{a^{3}}{3R}+(a^{2}-b^{2}{)}^{\frac{1}{2}}\left[\frac{mb}{tan\theta }+\frac{a^{2}-b^{2}}{3R}\right]\right),\;\delta -\frac{a}{R}\left[a-(a^{2}-b^{2}{)}^{\frac{1}{2}}\right]-\frac{na}{tan\theta }\left[\frac{\pi }{2}-{sin}^{-1}\left(\frac{b}{a}\right)\right]=0,$$
where *b* = *Rcosθ*; *a* is the contact radius; *R* is the radius of the spherical tip; *m* = 2^1/2^/*π* and *n* = 2^3/2^/*π* for the top of the pyramid; and *m* = 1/2 and *n* = 1 for the cone.

Based on experimental data and simulations, R. Vargas-Pinto et al. concluded that the cortical layer is examined with sharp probes, while the rigidity of the cortical layer and cytoskeleton is recorded with spherical probes [58]. Work hardening or the influence of a solid substrate is unlikely due to the increase in the Young’s modulus with increasing indentation value; therefore, the presence of the cortical layer directly affects the discrepancy between the results obtained with a spherical or sharp probe. However, in the presented model, the elastic component and the active stress component are combined into an effective elastic response for ease of calculation. Accounting for these components would make it possible to characterize the mechanical characteristics of individual cell structures in more detail.

The main limitation of the use of the Hertz model in the study of biological objects is the impossibility of studying the viscoelastic properties, as in this model the curves of the approach and withdrawal forces coincide (i.e., there is no hysteresis). As the viscoelasticity of the sample is the main source of hysteresis in liquid media, the hysteresis area of the force curve is used to evaluate the viscoelastic properties of cells.

Ting’s viscoelastic solution reflects the approach–retraction hysteresis well, although it requires an appropriate choice of the viscoelastic function. In a recent work, Y. Efremov et al. presented a method based on the principle of elastic viscoelasticity (Figure 3G) and confirmed its applicability using FEM and experiments on living cells and hydrogels with known mechanical characteristics [61]. Equations (35) and (36) describe Ting’s solution for indentation of a viscoelastic sample with a rigid spherical probe tip:(35)$$F\left(t,\delta \left(t\right)\right)=\left\{\begin{array}{c} \frac{4\sqrt{R}}{3(1-v^{2})}\int_{0}^{t}E\left(t-\xi \right)\frac{{\partial \delta }^{\frac{3}{2}}}{\partial \xi }d\xi ,\;\;\;\;\;\;0\leq t\leq t_{m}\\ \frac{4\sqrt{R}}{3(1-v^{2})}\int_{0}^{t_{1}(t)}E\left(t-\xi \right)\frac{{\partial \delta }^{\frac{3}{2}}}{\partial \xi }d\xi ,\;\;\;\;\;\;t_{m}\leq t\leq t_{ind} \end{array}\right.\;\;,\;\;\;$$
(36)$$\int_{t_{1}(t)}^{t}E\left(t-\xi \right)\frac{\partial \delta }{\partial \xi }d\xi =0,$$
where *F* is the force acting on the tip of the cantilever; *δ* is the indentation depth; *t* is the time initiated at the moment of initial contact (with *t_m_* being the duration of the rendezvous phase and *t_ind_* the duration of the full indentation cycle); *t*_1_ is an auxiliary function defined by Equation (36); ξ is a dummy time variable required for integration; *E(t)* is Young’s modulus of relaxation; *v* is Poisson’s ratio; and *R* is the radius of the indenter.

Y. Efremov et al. described the function of the relaxation modulus by rheological models (standard linear solid-state rheology and power-law rheology) as presented in Equations (37) and (38) (Figure 3G), respectively [61]. The standard linear rigid body model is a combination of a spring and damper, with the spring parallel to the Maxwell element. With the successive addition of a Maxwell element different from the existing one (spring to shock absorber or shock absorber to spring), a power-law rheology model is obtained in which there are several relaxation times:(37)$$E\left(t\right)=E_{\infty }+(E_{0}-E_{\infty })e^{-\frac{t}{\tau }},$$
(38)$$E\left(t\right)=E_{0}(1+\frac{t}{t^{\prime }}{)}^{-\alpha },$$
where *E*_0_ is the instantaneous modulus, E_∞_ is the long-term modulus, and τ is the relaxation time.

Another example of using a viscoelastic model with continuous relaxation spectra is presented by P. Cai et al. [62], where a stretched Kohlrausch–Williams–Watts exponential function was used for stepwise data analysis [86]:(39)$$E\left(t\right)=E_{\infty }+(E_{\infty }-E_{\infty })e^{-(\frac{t}{\tau_{r}}{)}^{\beta }}$$
where *β* is the exponent used to represent relaxation time dispersion processes in the system and *τ_r_* is the characteristic relaxation time.

In an earlier study by Y. Efremov et al., the Johnson–Kendall–Roberts model was applied to retraction curves to account for adhesion [63]. In the Johnson–Kendall–Roberts model (Figure 3H), the indentation depth *δ*, contact radius *a*, and maximum adhesive force *F_ad_* are represented by the following equations:(40)$$\delta =\frac{a^{2}}{R}-\frac{3}{4}\sqrt{\frac{aF_{ad}}{RK}};$$
(41)$$a=\sqrt{\left[\frac{R}{K}(\sqrt{F_{ad}}+\sqrt{F+F_{ad}}{)}^{2}\right]};$$
(42)$$F_{ad}=\frac{3}{2}\pi \gamma R,$$
where *K =* (4/3)*E*(1 − ν^2^) is the sample elasticity constant, *F* is the normal loading force, and *γ* is Dupre’s work of adhesion [65].

The Johnson–Kendall–Roberts model fits the retraction part well, with force–distance curves obtained at medium indentation speeds and low retention times to obtain data on the maximum adhesive force, Young’s modulus, and zero indentation point.

Summing up this section, we note the following conditions for choosing an analytical model. The Hertz model, which is the most widely used, does not provide for sticking of the probe to the sample; because of this, when choosing this model it is possible to obtain only the elastic properties of the object of study. In the theory of elastic shells, the main problem is the definition of boundary conditions. Moreover, due to the large thickness of the cell wall, its analysis using shell theory is not possible. While the core–shell model is well suited for studying the cell wall, it should be considered that the AFM method can only evaluate the outer layer of the cell, and this model does not consider the bottom effect. It is worth considering the discrepancy between the results obtained on spherical and sharp probes as well.

Despite significant progress in the understanding of cell mechanics, several provisions have not yet been studied. Combining AFM with other methods, such as scanning ion-conductance microscopy [87], could allow for obtaining data on mechanical properties that can effectively complement each other.

## 4. Mathematical Models for Measuring Mechanical Properties by the SICM Method

Recently, the use of the SICM method in the study of microbial cells, including yeast, has been developing [6,7,8,9]. The reason for this is the possibility of studying biological samples in their biological environment, as well as its low invasiveness. Scanning is non-contact and can provide the true topography of soft samples at a resolution comparable to AFM. Other advantages of the SICM method compared to AFM are the absence of lateral forces from the probe impacting on the sample and the reduction of the “height artifact” (10% underestimation of the height of a soft object by the SICM method versus 70% by the AFM method). The tilt angle of the nanocapillary used in the SICM method is smaller, which makes it possible to visualize details with almost vertical surface slopes. Due to its insensitivity to viscous drag forces, the imaging speed of SICM is higher than that of AFM [10]. The analytical models presented in this section are presented in Table 3.

The innovative SICM method uses a nanopipette as a scanning probe. The scanning system feeds back a constant ion current flowing through the nanopipette to approach the cell surface while maintaining a constant tip-to-surface distance approximately equal to the inner radius of the nanopipette. Based on the obtained capillary heights, a three-dimensional topographic image of the cell membrane is created near the sample [91] (Figure 4A). To obtain the mechanical properties of the cell, Rheinlaender, J. and Schäffer applied hydrostatic pressure through a nanopipette [89].

To convert the tangent of the curve (IZ-curves) of the fall of the ion current from the vertical position of the probes into the local stiffness of the sample in terms of Young’s modulus, the authors created a model based on finite element calculations. By simulating the fluid flow caused by a pressure *p*_0_ applied to the upper end of the pipette and calculating the resulting deformation of an elastic sample as a function of z, the authors obtained IZ curves and their s between 98% and 99% of the current for various *E/p*_0_ ratios. The empirically subject relationship between s and the Young’s modulus of the sample is described by the following equation:(43)$$E\left(s\right)=p_{0}A({\frac{S_{\infty }}{S}-1)}^{-1},$$
where *S_∞_* is the *s* for an infinitely rigid sample and *A* is a constant depending on the geometry of the pipette.

Previously, D. Sanchez et al. simulated hydrodynamics inside and outside the nanopipette tip [88]. Based on the Hagen–Poiseuille law, an equation was derived that relates the flow of ion current *I*_0_ through a nanopipette with the pressure drop Δ*P* of the liquid in the capillary and the environment:(44)$$I_{0}=\frac{3\pi tan\theta }{8\eta }r_{i}^{3}∆P,$$
where Δ*P* is the hydrostatic pressure drop between the nanopipette tip and the environment, *r_i_* is the radius of the inner hole of the nanopipette tip, *θ* is the semi-cone angle of the inner wall of the nanopipette tip, and *h* is the viscosity of the liquid.

D. Sanchez et al. used FEM to simulate the fluid flow in the nanopipette and the force at the tip boundary acting on a flat non-deformable sample [88]. They derived Equation (45) for the total normal force exerted on a flat surface of a non-deformable sample.
(45)$$F=2\pi \int_{0}^{\infty }P(r)rdr$$

At small distances between the nanopipette tip and the sample surface, the total impact force of the probe is greater, and depends on the distance from the tip to the surface (*z*) and the speed of approach of the probe (*v*). Thus, the tangential shear stress, presented in Equation (46), is determined by the viscosity *h* multiplied by the derivative with respect to the velocity component d*v_r_*/d*z* tangential to the surface; it is maximum at *r* = *r_i_* and is equal to 12 nN/mm^2^ [88].
(46)$$\frac{F}{A}=\frac{\eta dv_{r}}{dz}$$

Measurement of the mechanical properties of the cell membrane via the SICM method is possible by applying hydrostatic pressure as well as by indentation via internal colloidal pressure between the cell surface and the surface of the nanopipette tip [92]. R. Clarke et al. presented a new approach to the study of the Young’s modulus of mammalian cells [90]. The self-voltage in terms of the ion current drop Δ*I* as the probe approaches the sample is presented in Equation (47):(47)$$\sigma =H/6\pi (\left(\frac{r}{x}\right)ln(I_{0}/\Delta I){)}^{3},$$
where *H* is Hamaker’s constant, *r* is the nanopipette radius, *x* is an empirically determined constant equal to 3.6 ± 0.2, I_0_ is the ion current far from the sample surface, and Δ*I* is the drop in the ion current when approaching the sample.

Thus, for cells without a glycocalyx, the Young’s modulus is
(48)$$E=\frac{\sigma }{\left(1-\frac{h}{h_{0}}\right)},$$
where *σ* is the stress from Equation (47), *h*_0_ is the cell height, and *h* is the height of the indented cell area.

For cells with a glycocalyx, the total Young’s modulus is
(49)$$E=(h_{c}+h_{s})/(\left(\frac{h_{c}}{E_{c}}\right)+(\frac{h_{c}}{E_{c}})),$$
where *h_c_* is the height of the cell cortex, *h_s_* is the height of the soft area of the cell cytoskeleton, *E_c_* is the Young’s modulus of the cell cortex, and *E_s_* is the Young’s modulus of the cell cytoskeleton.

The most relevant technique for measuring the Young’s modulus via the SICM method has been presented by Kolmogorov et al. [93] based on the deformation of a double electric layer of decan–saline solution from a nanopipette [90]. The surface deformation is calculated from the dependence of the ion current on the distance to the sample surface and the dependence of the ion current on the distance to the undeformed surface (Figure 4B). The internal force is calculated from the force balance equation:(50)$$\vec{F}=2\vec{F_{\sigma }},$$
where *F_σ_* is the surface tension force of the decan–salt layer.

In the described work, it is assumed that the deformed region has the shape of a sphere with a contact radius *a*; according to the Hertz model, this parameter is expressed by Equation (51). The surface tension force is calculated using the Laplace Equation (52), in which *R* is the inner radius of the nanopipette, *a* is the distance between the point of maximum indentation depth (*d*) and the point of zero indentation of the surface, and *σ* is the surface tension parameter. From the combination of Equations (50)–(52), the internal force is calculated according to Equation (53).
(51)$$a=\sqrt{R*d},$$
(52)$$F_{\sigma }=2\pi a\sigma ,$$
(53)$$F=4\pi \sigma \sqrt{R*d}.$$

Summing up, the considered methods demonstrate the possibility of mapping the mechanical properties of mammalian cells with high resolution. In the case of AFM, it is possible to study the viscoelastic properties of biological objects; however, when studying soft cells this method is subject to scanning artifacts associated with large impact forces on the sample [8]. The SICM method in the no-pressure mode is free from this disadvantage, and the displacement from the tip surface to the cell surface is minimized. Unfortunately, at the moment there is no SICM method for obtaining viscoelastic properties. In this connection, the combination of these two methods has the potential to expand the study of numerous processes in the field of biophysics.

## 5. Conclusions

The correct choice of an analytical model for measuring the mechanical properties of cells plays a key role in obtaining relevant parameters. Despite the large amount of experimental data obtained, the choice of an appropriate mechanical model for a particular biological sample remains a matter of debate. The main reason for this is the complexity of the cell structure as well as the impact of third-party factors or instrumental impacts on the sample. The studies presented in this review can provide an approximate idea of the currently available models used to study the elastic and viscoelastic properties of cells. The currently available SICM method for determining cell stiffness does not accurately reflect the properties of the yeast cell wall. Familiarization with the generalized models presented in this work can allow the choice of the most appropriate cell model in the SICM method, which can contribute to progress in obtaining the elastic modulus of yeasts and of various structures on their surface. Moreover, the combination of SICM with AFM can become an advanced tool in the development and testing of mechanical models of cells, which can later be used in the development of antifungal drugs or the evaluation of antimicrobial therapies. In the near future, we plan to modernize the SICM method based on the presented models in order to obtain the viscoelastic properties of biological objects, including yeast cells.

## Figures and Tables

**Figure 1 cells-12-01946-f001:**
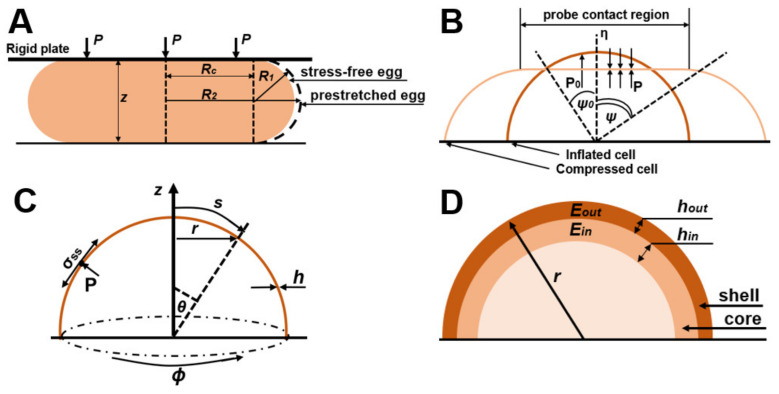
(**A**) Illustration of a sea urchin model, showing a yeast cell when compressed between two large rigid plates. (**B**) Illustration of a model of a hollow sphere filled with gas; cell geometry: *ψ*_0_ is the angular position of a point on the cell wall from the vertical axis of symmetry before compression and *ψ* determines the angle of a point on the edge of the contact area between the compression surface and the cell after compression. (**C**) Illustration of the shell model showing the geometry of the system and the image of the increase in the viscosity of the cell wall. (**D**) Illustration of the core–shell model; normalized compression force with total wall stiffness *F/r(Eh)_tota_*_l_ under parallel compression for a two-layer model with an inner wall *h_in_/r* = 0.01 and an outer wall *h_out_/r* = 0.04 with different ratios of Young’s moduli.

**Figure 2 cells-12-01946-f002:**
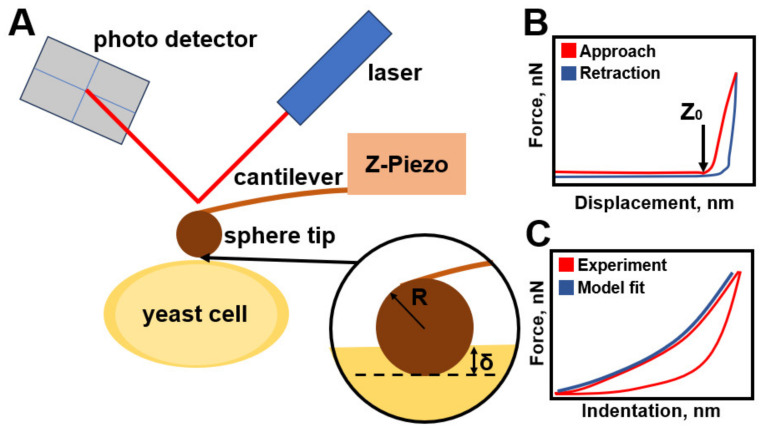
(**A**) Schematic representation of the elements of the AFM setup. The spherical tip interacts with the surface of the sample, which leads to the deflection of the microcantilever, which is recorded by the photodetector using a laser beam reflected from the microcantilever. In a typical measurement of mechanical curves, the base of the micro console is approached or retracted at a constant vertical speed and the force is recorded. (**B**) An example of the resulting F-Z curves, in which the arrow indicates the point of contact of the probe with the sample. (**C**) An example of converting an F-Z curve to a force–indentation curve, from which Young’s modulus is obtained.

**Figure 3 cells-12-01946-f003:**
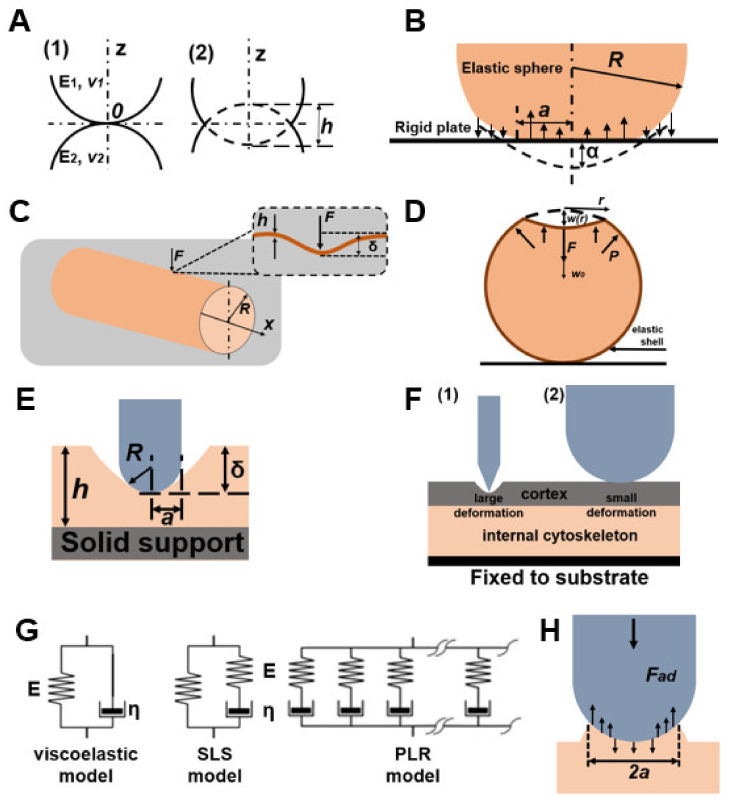
(**A**) Scheme of the Hertz model, where E is Young’s modulus, v is Poisson’s ratio, and X is indentation. Indexes 1 and 2 refer to two bodies, respectively. (**B**) Scheme of the DMT model, where R is the radius of the elastic sphere, *a* is the contact radius, and α is the shift from the center of the elastic sphere. (**C**) Diagram of a cylindrical shell model, where the AFM cantilever tip applies a normal force F, indenting (δ) a hypha with internal pressure P, radius R, and thickness *h*. (**D**) Illustration of the elastic shell model. A spherical shell with thickness h and undeformed radius R experiences internal pressure P and is loaded with a vertical point force F at the pole. This causes a vertical deflection *w(r)* and, in particular, a displacement *w*(0) = −w_0_ at the point of application of the force. (**E**) Illustration of the Non-Hertz Model. (**F**) Schematic of the FEM for sharp tip (1) or spherical tip (2). (**G**) Viscoelastic models can be used to relate the stress and strain by using elastic (springs, denoted E) and viscous (dashpots, denoted η) parts. (**H**) Scheme of the Johnson–Kendall–Roberts model, where *a* is the contact radius and *F_ad_* is the maximum adhesive force.

**Figure 4 cells-12-01946-f004:**
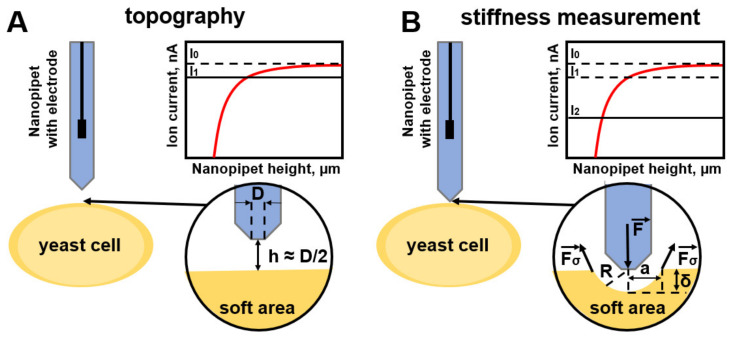
(**A**) Schematic representation of obtaining topography via the SICM method, where the probe stops when the ion current drops by 0.5% while being approximately at the nanocapillary radius from the sample surface. (**B**) Schematic representation of the measurement of sample stiffness by indenting it due to the internal colloidal pressure of the nanocapillary; the ion current drop is 2%.

**Table 1 cells-12-01946-t001:** Analytical models applicable to micromanipulation techniques.

Source	Model	Function	An Object	Description
J.D. Stenson et al. [31,35]	Sea urchin egg model	Infinitely small deformation in Equations (10)–(12), final deformation in Equations (13)–(15), Hankey’s deformation in Equations (16)–(18)	yeast cell wall	In the model, cells are thin-walled, liquid-filled spheres; the desired characteristics depend on the Poisson’s ratio and the thickness of the cell wall. It is possible to neglect cell wall permeability at high strain rates. Fixing the initial stretch factor leads to an inaccurate estimate of the elastic modulus.
Feng and Yang [36]	Model of compression of hollow spheres filled with gas	Equations (4)–(7) constitutive equations for contact and non-contact regions	cell wall of tomato cells	The cell wall in this model is divided into areas in contact and areas not in contact with compressive forces.
Banavar et al. [37]	Shell theory	Local normal balance of forces of the cell wall in Equation (19) Stresses in the cell wall according to Equations (20) and (21)	growing cell wall dynamics	The growing cell wall behaves like an inhomogeneous viscous liquid with a spatially changing viscosity that increases with distance from the growth apex
Mercade’-Prieto et al. [38]	Core-shell model	wall stiffness *F/r(Eh)_out_*	cell wall	The model gives an estimate of the overall stiffness of the cell wall (Figure 1D).

**Table 2 cells-12-01946-t002:** Analytical models applicable to the AFM method.

Source	Model	Function	An Object	Description
H. Hertz [52]	Hertz Model	Cantilever Force Equation (22), effective Young’s modulus Equation (23) (When the material of the tip is significantly harder than the material of the sample, Equation (24))	homogeneous smooth bodies	The model is used under the assumptions that the indenter shape is parabolic, and the sample thickness is much greater than the indentation depth. The model does not allow the probe to stick to the sample.
B. Derjaguin [53]	DTM model		cell wall	The model is applicable in the presence of long-range surface forces outside the area of contact between the probe and the sample and is valid in the event of weak adhesion between the nanoindenter and the outer surface of the sample. Its use is a priority for objects with low cohesion and a small radius of curvature.
Zhao et al. [46]	Cylindrical shell model	The modulus of elasticity of the cell wall in Equation (28)	cell wall	In the technique, F and δ are linearly dependent on each other, while the cell wall elasticity constant kw depends on the mechanical properties and dimensions of the cell wall but does not depend on the internal pressure of the cell.
Vella et al. [54]	Elastic shell model		internal pressure in yeast cells	Young’s modulus is an order of magnitude higher than the values obtained using the Hertz model.
Mercade’-Prieto et al. [38]	Single layer sphere	The values of F and Eh are calculated from Equations (28) and (29)	cell wall	Corrected values of the Young’s modulus are higher than using Hertz–Sneddon analysis but lower than using micromanipulation compression.
Mercade’-Prieto et al. [38]	Double layer model	Force profile at small deformations in Equation (31).	cell wall	The model of a two-layer cell wall suggests the possibility of estimating the elastic modulus by AFM only for the outer layer.
E. A-Hassan, S.P. Timoshenko [55,56]	Theory of elastic shells	Young’s modulus is estimated from the ratio between the effective Young’s modulus, shell thickness and bending modulus	cells	Cells in the model are represented as shells filled with liquid.
P. Garcia & R. Garcia [57]	Non-Hertz model	In the case of a paraboloid probe, the force is expressed by Equation (33).	mammalian cells attached to a solid support	The cell’s Young’s modulus depends on the solid substrate, and the bottom effect artifact is determined by the ratio between the contact radius and cell thickness. The model is applicable when the indentation is less than or equal to the tip radius.
R. Vargas-Pinto et al. [58]	Hertz Model and Contact Model	The force, in the case of a spherical tip, is expressed by Equation (29) In the case of a sharp tip, the model is used Rico et al. [59] and Briscoe et al. [60], where the force is expressed in Equation (34)	mammalian cells with cortex	Combining the models resolved the issue of inaccuracy in determining the rigidity of the cage. Sharp probes examine the cortical layer, and spherical probes record the rigidity of the cortical layer together with the cytoskeleton. In the model presented, the elastic component and the active stress component are combined into an effective elastic response for ease of calculation.
Y. Efremov et al. [61]	Elastic-Viscoelastic Compliance	Ting’s solution for indentation of a viscoelastic sample with a rigid spherical tip Equations (35) and (36).	living cells and hydrogels	It reflects the approach-retraction hysteresis well but requires an appropriate choice of the viscoelastic function.
Y. Efremov et al. [61] P. Cai et al. [62]	Standard Linear Solid-State Rheology and Power Rheology	Relaxation time Equations (35) and (36), Kohlrausch–Williams–Watts function Equation (37).	living cells	The standard linear rigid body model is a combination of a spring and damper, in which the spring is parallel to the Maxwell element.
Y. Efremov et al. [63]	Johnson-Kendall-Roberts model	The indentation depth, contact radius, and maximum adhesive force are presented in Equations (38)–(40), respectively)	living cells and hydrogels	The model fits the retraction part well with force-distance curves.

**Table 3 cells-12-01946-t003:** Analytical models applicable to the SICM method.

Source	Model	Function	An Object	Description
D. Sanchez et al. [88], Rheinlaender, J., & Schäffer [89]	Hydrodynamic model	The force exerted on a flat surface in Equation (43). Young’s modulus of the sample in Equation (41).	cell membrane	To obtain the mechanical properties of the cell, hydrostatic pressure is applied through a nanopipette, which can lead to a mechanical response of the cell.
R. Clarke et al. [90]	Internal colloidal pressure model	The modulus of elasticity of the cell wall in Equation (28)	cells with glycocalyx and cells without glycocalyx	Indentation is performed by means of internal colloidal pressure between the cell surface and the surface of the nanopipette tip, which significantly reduces the invasiveness of the method.
Kolmogorov et al. [61], Savin N. et al. [8]	Hertz Model	The internal force is presented in Equation (51).	Mammalian cells [61], yeast cells [8]	The technique is based on the deformation of a double electric layer of decan-saline solution with a nanopipette. The displacement from the tip surface to the cell surface is minimized. However, there is no method for obtaining viscoelastic properties in all presented SICM models.

## Data Availability

Not applicable.

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
