# Peer review of "Analytical Models for Measuring the Mechanical Properties of Yeast"

_cells, 2023, doi:10.3390/cells12151946_

Round 1

Reviewer 1 Report

The authors presented brief descriptions and final expressions of selected parameters derived from previously published theoretical models considering mechanical properties of yeast cells. Collection of derived equations can be of use in selection of an appropriate model to interpret experimental data. I suggest that the authors consider the comments below.

The authors claim: “This work may make it possible to select the most suitable functions for this method…” and present sets of final equations of various models. However, the reader cannot understand the derived expressions without acknowledging the model assumptions in detail as well as deriving the equations. Therefore I think that the claim of the authors should be rephrased. The readers may browse through the presented equations to see whether their experiments provide the input parameters, however, the authors should suggest in the text that in order to use the equations, the model should be thoroughly studied by following the original publication. I do not favour using the equations without understanding their foundations and limitations.  Also, please give some introductory explanation to the reader what is the sense of presenting the equations in such way (without derivations).

As many different models are mentioned and listed in the manuscript there are many parameters introduced; in particular, it is difficult to comprehend the model without having an idea how the system looks like (for each model). I suggest that the authors present illustrations to explain the model parameters and assumptions. If they feel that this would make the manuscript too long they can assemble supplementary material. But, please give a description of the system and basic assumptions for the models mentioned before presenting the equations.

I suggest that the tables with informations on the models are presented at the beginning of the respective sections (and not at the end).

Description of the models, starting line 67. The expression “modelling of compression in hollow spheres filled with gas and incompressible liquid, respectively” seems unclear to me. Does it mean that one model of yeast is a sphere filled with gas? And another model of yeast is a sphere filled with liquid? Please clarify.

I am not familiar with the term “mechanical cell membrane”. Please be more precise in a physical point of view (e.g. thin elastic shell or similar).

Then authors say that stresses are expressed as a wall tension and that and the wall cannot withstand out-of-plane shear stresses or bending moments. Also this seem unclear. It would be best if the authors stated which elements of the stress tensor were neglected. Also further on, the authors refer to Cauchy stresses and present two principal stresses (Eqs.(8) and (9)). While the detailed definitions of the system and basic assumptions of models are  expected to be given in the reviewed references, some description should be provided in order to enable the readers to understand the equations. The simplification of the general form of the Cauchy tensor should be explained, etc. Some illustration of the system would be of help.  Furthermore, the authors use the expressions such as Piola-Kirchoff stress, Green’s strain, Hank’s deformation, Hank’s deformity, ultimate deformation, etc. Please provide explanation and a reference for each of such expressions.

After Eq.(1) the authors describe e as “strain”, then they claim that “infinitesimal deformation”  in Eq.(1) was replaced…Please clarify.  Please explain what are E and H in more detail and justify the replacements.

Lines 92-110. Please clearly state the relations between the models [15], [17] and [20].  It seems as the authors wish to combine the models.

Eq.(19), Line 152: the illustration of the coordinate system would be of help to understand what are s, j and radius of the cell wall.

Line 175: what do the two bodies refer to?

Line 242: please describe what are F and Eh.

Line 249: what do the two layers correspond to?

Eq.(31): are 0,5 and 1,5 exponentials or indexes? It is unusual to give exponentials in decimals. Also, in English language the decimals are given by dots.  Please check the expression EpHe1,5.  It seems to me as pHe1,5  is a subscript, therefore it is an index?

Please explain the meaning of the symbols in Eqs. (32) and (33).

Line 411. It should be “h is the viscosity”

In some places, the use of the language should be more precise (e.g. mechanical cell membrane), however, this refers to the clarity of physical content and not grammatics or syntax.

Author Response

Dear reviewer, thank you for your report and questions.

Major paper changes.

  1. Figures have been added.

Point 1: The authors claim: “This work may make it possible to select the most suitable functions for this method…” and present sets of final equations of various models. However, the reader cannot understand the derived expressions without acknowledging the model assumptions in detail as well as deriving the equations. Therefore I think that the claim of the authors should be rephrased. The readers may browse through the presented equations to see whether their experiments provide the input parameters, however, the authors should suggest in the text that in order to use the equations, the model should be thoroughly studied by following the original publication. I do not favour using the equations without understanding their foundations and limitations.  Also, please give some introductory explanation to the reader what is the sense of presenting the equations in such way (without derivations).

Response 1: The claim has been rephrased. Added a warning to the reader in the introduction.

Point 2: As many different models are mentioned and listed in the manuscript there are many parameters introduced; in particular, it is difficult to comprehend the model without having an idea how the system looks like (for each model). I suggest that the authors present illustrations to explain the model parameters and assumptions. If they feel that this would make the manuscript too long they can assemble supplementary material. But, please give a description of the system and basic assumptions for the models mentioned before presenting the equations.

Response 2: Schematic images have been added in accordance with the original publications.

Point 3: I suggest that the tables with informations on the models are presented at the beginning of the respective sections (and not at the end).

Response 3: The position of the tables has been changed.

Point 4: Description of the models, starting line 67. The expression “modelling of compression in hollow spheres filled with gas and incompressible liquid, respectively” seems unclear to me. Does it mean that one model of yeast is a sphere filled with gas? And another model of yeast is a sphere filled with liquid? Please clarify.

Response 4: This refers to models of spheres, one of which is filled with gas and the other with an incompressible liquid. Clarification was added to the text of the publication.

Point 5: I am not familiar with the term “mechanical cell membrane”. Please be more precise in a physical point of view (e.g. thin elastic shell or similar).

Response 5: Has been corrected.

Point 6: Then authors say that stresses are expressed as a wall tension and that and the wall cannot withstand out-of-plane shear stresses or bending moments. Also this seem unclear. It would be best if the authors stated which elements of the stress tensor were neglected. Also further on, the authors refer to Cauchy stresses and present two principal stresses (Eqs.(8) and (9)). While the detailed definitions of the system and basic assumptions of models are expected to be given in the reviewed references, some description should be provided in order to enable the readers to understand the equations. The simplification of the general form of the Cauchy tensor should be explained, etc. Some illustration of the system would be of help.  Furthermore, the authors use the expressions such as Piola-Kirchoff stress, Green’s strain, Hank’s deformation, Hank’s deformity, ultimate deformation, etc. Please provide explanation and a reference for each of such expressions.

Response 6: It is indicated which tensor is neglected in the plane stress system. Descriptions of Cauchy stresses have been given. We have made speech errors in the presented physical terms. The misunderstanding has been corrected.

Point 7: After Eq.(1) the authors describe e as “strain”, then they claim that “infinitesimal deformation”  in Eq.(1) was replaced…Please clarify.  Please explain what are E and H in more detail and justify the replacements.

Response 7: An explanation has been added.

Point 8 Lines 92-110. Please clearly state the relations between the models [15], [17] and [20].  It seems as the authors wish to combine the models.

Response 8: Stenson et al. [20] developed a model based on Cheng, L.Y. et al. [17] functions. Feng and Yang [15] is a separate model. Explanation added to the text.

Point 9: Eq.(19), Line 152: the illustration of the coordinate system would be of help to understand what are s, j and radius of the cell wall.

Response 9: An illustration is shown in Figure 1 (C).

Point 10: Line 175: what do the two bodies refer to?

Response 10: Accompanied by Figure 3 (A).

Point 11: Line 242: please describe what are F and Eh.

Response 11: Has been described.

Point 12: Line 249: what do the two layers correspond to?

Response 12: The outer layer corresponds to the shell; the inner layer corresponds to the core. An explanation is included in the text.

Point 13: Eq.(31): are 0,5 and 1,5 exponentials or indexes? It is unusual to give exponentials in decimals. Also, in English language the decimals are given by dots.  Please check the expression EpHe1,5.  It seems to me as pHe1,5 is a subscript, therefore it is an index?

Response 13: 0.5 and 1.5 are indices, the equation has been corrected.

Point 14: Please explain the meaning of the symbols in Eqs. (32) and (33).

Response 14: The symbols in the equations have been clarified.

Point 15: Line 411. It should be “h is the viscosity”

Response 15: Corrected.

Reviewer 2 Report

The manuscript titled “Analytical Models for Measuring the Mechanical Properties of Yeast” by Savin, N.; et al. is a review work where the authors show the promising capabilities of atomic force microscopy (AFM) and scanning ion-conductance microscopy (SICM) techniques to address the mechanical properties with special focus on yeasts. The authors describe the existing models for AFM and SICM, respectively and when is more appropiate their use. The paper is interesting and the division of the subsequent sections is well-designed.

However, it exists some points that need to be addressed (please, see them below detailed point-by-point). The most relevant outcomes depicted by the authors can contribute to better understand a current topic of growing interest. The interrogation of mechanical properties at the single cellular level can open new gates to unravel the impact of antifungal compounds on the viability of yeast membranes. Furthermore, the knowledge shown in this paper can be extrapolated not only in yeasts, but also for many other cellular systems. For this reason, I will recommend the present scientific manuscript for further publication in Cells once all the below described suggestions will be properly fixed.

Here, there exists some points that must be covered in order to improve the scientific quality of the manuscript paper:

1) INTRODUCTION. “At the moment, (…) most advance method (…) is probe microscopy” (lines 35-37). Please, the authors should modify this sentence by “(…) most advanced methods (…) are single probe microscopies (SPMs)”.

2) “which has a high spatial resolution (…) of the AFM [5] and SICM [6-9] methods” (lines 37-39). Here, the authors should also point out the advantages offered by these techniques like the possibility to conduct measurements in liquid media mimicking the yeast intracellular conditions. Moreover, the terms “atomic force microscopy” and “scanning ion-conductance microscopy”should be full named the first time that they appear in the main manuscript body text. Finally, the authors should list the potential limitations related to AFM and SICM techniques (as the requirement to attach the sample on solid surfaces) and compared the resolution achieved for both techniques [1].

[1] Rheinlaender, J.; et al. Comparison of scanning ion conductance microscopy with atomic force microscopy for cell imaging. Langmuir 2011, 27, 697-704. https://doi.org/10.1021/la103272y.

3) Furthermore, in the aforementioned statement (lines 37-39) the authors should also briefly discuss about other alternative techniques to extract the mechanical properties of the tested sample as parallel-rheology [2] or optical tweezers [3], respectively.

[2] Bonfanti, A.; et al. A unified rheological model for cells and cellularised materials. R. Soc. Open Sci20207, 190920. https://doi.org/10.1098/rsos.190920.

[3] Magazzù, A.; et al. Investigation of Soft Matter Nanomechanics by Atomic Force Microscopy and Optical Tweezers: A comprehensive Review. Nanomaterials 2023, 13, 963. https://doi.org/10.3390/nano13060963.

4) The authors should also catalogue other physico-chemical properties that can be simultaneously recorded with the mechanical parameters like the tip-sample adhesion [4] or the energy dissipation [5] in the case of AFM measurements.

[4] Lostao, A.; et al. Recent advances in sensing the inter-biomolecular interactions at the nanoscale – A comprehensive review of AFM-based force spectroscopy. Int. J. Biol. Macromol. 2023, 238, 124089. https://doi.org/10.1016/j.ijbiomac.2023.124089.

[5] Pukhova, V.; et al. Energy dissipation in multifrequency atomic force microscopy. Beilstein J. Nanotechnol. 2014, 5, 494-500. https://doi.org/10.3762/bjnano.5.57.

5) MATHEMATICAL MODELS FOR MEASURING MECHANICAL PROPERTIES BY MICROMANIPULATION. “For example, some models (…) which includes the Poisson ratio” (lines 84-86). What value of Poisson ratio is suggested to be used for yeast samples? The authors should briefly indicate this information.

6) MATHEMATICAL MODELS FOR MEASURING MECHANICAL PROPERTIES BY THE AFM METHOD. Table 2 (line 374). The authors should also add the Derjaguin-Muller-Toporov (DMT) model which has been successfully used to determine the Young’s modulus of native yeast cells and under chemical stress conditions [6].

[6] Ribeiro, R.A.; et al. Crosstalk between Yeast Cell Plasma Membrane Ergosterol Content and Cell Wall Stiffness under Acetic Acid Stress Involving Pdr18. J. Fungi 2022, 8, 103. https://doi.org/10.3390/jof8020103.

Furthermore, the authors should state that DMT model is appropiate when there exist long-range surface forces outside the tip-sample contact area. Thus, this model is valid when weak adhesion events appear between the nanoindenter and the external sample surface.

7) The authors should provide a schematic representation of AFM and SICM techniques to explain the working principles when these tools are used to determine the mechanical properties of the sample of interest. This is a crucial point to significantly aid to potential readers to better understand the relevance of this work.

8) MATHEMATICAL MODELS FOR MEASURING THE MECHANICAL PROPERTIES BY THE SICM METHOD. This section is clear and accurate. No actions are requested of the authors.

9) DISCUSSION. It may be convenient to highlight some future avenues that the topic covered in this work could address in the near future.

10) REFERENCES. The references are in the proper format style of Cells. No actions are requested of the authors (with exception of reference number 6 where the journal name should appear in abbreviated form).

Minor editing of English language required

Author Response

Dear reviewer, thank you for your report and questions.

Major paper changes.

  1. Figures have been added.

Point 1: NTRODUCTION. “At the moment, (…) most advance method (…) is probe microscopy” (lines 35-37). Please, the authors should modify this sentence by “(…) most advanced methods (…) are single probe microscopies (SPMs)”.

Response 1: Corrected.

Point 2: “which has a high spatial resolution (…) of the AFM [5] and SICM [6-9] methods” (lines 37-39). Here, the authors should also point out the advantages offered by these techniques like the possibility to conduct measurements in liquid media mimicking the yeast intracellular conditions. Moreover, the terms “atomic force microscopy” and “scanning ion-conductance microscopy”should be full named the first time that they appear in the main manuscript body text. Finally, the authors should list the potential limitations related to AFM and SICM techniques (as the requirement to attach the sample on solid surfaces) and compared the resolution achieved for both techniques [1].

Response 2: Information has been added.

Point 3: Furthermore, in the aforementioned statement (lines 37-39) the authors should also briefly discuss about other alternative techniques to extract the mechanical properties of the tested sample as parallel-rheology [2] or optical tweezers [3], respectively.

Response 3: Information has been added.

Point 4: The authors should also catalogue other physico-chemical properties that can be simultaneously recorded with the mechanical parameters like the tip-sample adhesion [4] or the energy dissipation [5] in the case of AFM measurements.

Response 4: Information has been added.

Point 5: MATHEMATICAL MODELS FOR MEASURING MECHANICAL PROPERTIES BY MICROMANIPULATION. “For example, some models (…) which includes the Poisson ratio” (lines 84-86). What value of Poisson ratio is suggested to be used for yeast samples? The authors should briefly indicate this information.

Response 5: Information has been added.

Point 6: MATHEMATICAL MODELS FOR MEASURING MECHANICAL PROPERTIES BY THE AFM METHOD. Table 2 (line 374). The authors should also add the Derjaguin-Muller-Toporov (DMT) model which has been successfully used to determine the Young’s modulus of native yeast cells and under chemical stress conditions [6].

Furthermore, the authors should state that DMT model is appropiate when there exist long-range surface forces outside the tip-sample contact area. Thus, this model is valid when weak adhesion events appear between the nanoindenter and the external sample surface.

Response 6: The model has been added.

Point 7: The authors should provide a schematic representation of AFM and SICM techniques to explain the working principles when these tools are used to determine the mechanical properties of the sample of interest. This is a crucial point to significantly aid to potential readers to better understand the relevance of this work.

Response 7: A schematic image has been added.

Point 8: MATHEMATICAL MODELS FOR MEASURING THE MECHANICAL PROPERTIES BY THE SICM METHOD. This section is clear and accurate. No actions are requested of the authors.

Point 9: DISCUSSION. It may be convenient to highlight some future avenues that the topic covered in this work could address in the near future.

Response 9: The conclusions have been slightly modified.

Point 10: REFERENCES. The references are in the proper format style of Cells. No actions are requested of the authors (with exception of reference number 6 where the journal name should appear in abbreviated form).

Response 10: The reference has been corrected.

Round 2

Reviewer 1 Report

I have read the improved version and think that it is appropriate for publication.

It would be optimal to have minor improvements of the English language in some places.